# ELFS: Label-Free Coreset Selection with Proxy Training Dynamics

**Haizhong Zheng**[*][1], **Elisa Tsai**[*][1], **Yifu Lu**[1], **Jiachen Sun**[1],
**Brian R. Bartoldson**[2], **Bhavya Kailkhura**[2], **Atul Prakash**[1]

[1]University of Michigan
[2]Lawrence Livermore National Laboratory
`{hzzheng,eltsai,yifulu,jiachens, aprakash}@umich.edu`
`{bartoldson,kailkhura1}@llnl.gov`

## Abstract

High-quality human-annotated data is crucial for modern deep learning pipelines, yet the human annotation process is both costly and time-consuming. Given a constrained human labeling budget, selecting an informative and representative data subset for labeling can significantly reduce human annotation effort. Well-performing state-of-the-art (SOTA) coreset selection methods require ground truth labels over the whole dataset, failing to reduce the human labeling burden. Meanwhile, SOTA label-free coreset selection methods deliver inferior performance due to poor geometry-based difficulty scores. In this paper, we introduce *ELFS* (Effective Label-Free Coreset Selection), a novel label-free coreset selection method. *ELFS* significantly improves label-free coreset selection by addressing two challenges: 1) *ELFS* utilizes deep clustering to estimate training dynamics-based data difficulty scores without ground truth labels; 2) Pseudo-labels introduce a distribution shift in the data difficulty scores, and we propose a simple but effective double-end pruning method to mitigate bias on calculated scores. We evaluate *ELFS* on four vision benchmarks and show that, given the same vision encoder, *ELFS* consistently outperforms SOTA label-free baselines. For instance, when using SwAV as the encoder, *ELFS* outperforms D2 by up to 10.2% in accuracy on ImageNet-1K. We make our code publicly available on GitHub[1].

## 1 Introduction

Modern machine learning systems, particularly deep learning frameworks, are increasingly data-intensive and computationally demanding (Touvron et al., 2023; Achiam et al., 2023). High-quality labeled data is crucial in the deep learning pipeline. Given the same model architecture, improving the quality of training data can significantly boost model performance (Zheng et al., 2023; Maharana et al., 2023; Xia et al., 2023; Choi et al., 2023). However, generating high-quality labeled data typically requires costly human annotation efforts. Selecting an

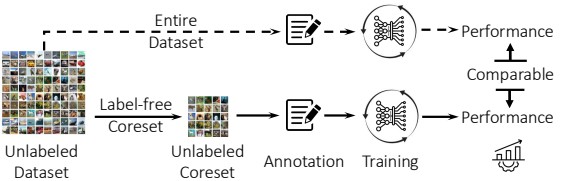

Figure 1: Label-free Coreset Selection Scheme. The goal of label-free coreset selection is to identify an informative and representative subset of the data without relying on ground truth labels, minimizing human annotation efforts in deep learning pipelines.

informative and representative subset for labeling can substantially reduce these costs (Bartoldson et al., 2023), thereby optimizing the use of resources and enhancing the overall efficiency of the data annotation process.

State-of-the-art (SOTA) coreset selection methods use data difficulty scores derived from training dynamics to select coresets (Pleiss et al., 2020; Toneva et al., 2018; Paul et al., 2021; Zheng et al.,

---

[*]Equal contribution
[1]https://github.com/eltsai/elfs

2023; Maharana et al., 2023; Xia et al., 2023; Choi et al., 2023). However, these methods require ground truth labels over the whole dataset to calculate difficulty scores, which fails to achieve the purpose of reducing human annotation efforts. Existing label-free coreset selection methods calculate importance scores based on geometric properties in the embedding space (Sorscher et al., 2022a; Maharana et al., 2023). However, compared to training dynamics scores, these geometry-based scores often fail to accurately estimate data difficulty and lead to poor coreset selection performance—even often worse than random sampling at some pruning rates (Maharana et al., 2023). To utilize more accurate scores calculated by training dynamics in the label-free coreset selection setting, there are two primary challenges that need to be solved:

1. *How to estimate training dynamics without ground truth labels?*
2. *How to select high-quality coresets with proxy training dynamics?*

In this paper, we propose a novel coreset selection algorithm, *ELFS* (Effective Label-Free Coreset Selection) to enhance label-free coreset selection by addressing those two challenges. For the *first challenge*, we show that pseudo-labels provide a good estimation for training dynamics-based difficulty scores. More specifically, ELFS utilizes deep clustering (Van Gansbeke et al., 2020; Adaloglou et al., 2023) to assign a pseudo-label to each data point. We empirically show that training dynamic scores derived from pseudo-labels provide a good proxy to measure data difficulty for coreset selection (Section 4.3). For the *second challenge*, we observe that directly applying existing selection methods like CCS (Zheng et al., 2023) results in a performance gap when compared to coreset selection with ground truth labels. Our further analysis shows that this gap is due to the distribution shift in the difficulty scores caused by inaccurate pseudo-labels. This distribution shift leads traditional sampling methods to select more easy examples, resulting in inferior performance (Section 4.3). To mitigate this distribution shift issue, we propose a simple yet effective double-end pruning method for the pseudo-label-based difficulty scores. This double-end pruning significantly reduces the number of selected easy examples and improves the coreset selection performance.

To verify the effectiveness of *ELFS*, we evaluate it on four vision benchmarks: CIFAR10, CIFAR100 (Krizhevsky et al., 2009), STL10 (Coates et al., 2011), and ImageNet-1K (Deng et al., 2009). The evaluation results show that, given the same vision encoder, *ELFS* consistently outperforms all label-free baselines on all datasets at different pruning rates. For instance, when using SwAV as the encoder, *ELFS* outperforms D2 by up to 10.2% in accuracy on ImageNet-1K. Similarly, when employing DINO as the encoder, *ELFS* outperforms FreeSel with an accuracy gain of up to 3.9% on ImageNet-1K. For some pruning rates (e.g., 30% and 50% for ImageNet-1K), *ELFS* even achieves comparable performance compared to the best supervised coreset selection performance. In addition to performance comparisons, we also conduct comprehensive ablation studies to analyze the contribution of each component of *ELFS*.

## 2 RELATED WORK

### 2.1 CORESET SELECTION

**Supervised Coreset Selection.** Coreset selection aims to select a representative subset of the training data in a one-shot manner, which can be used to train future models while maintaining high accuracy. Supervised coreset selection can be categorized into two primary categories (Guo et al., 2022). (1) *Optimization-based methods* aim to select subsets that produce gradients similar to the entire dataset (Mirzasoleiman et al., 2020; Killamsetty et al., 2021a). (2) *Difficulty score methods* select examples based on metrics such as forgetting scores (Toneva et al., 2018), area under the margin (AUM) (Pleiss et al., 2020), EL2N (Paul et al., 2021), and Entropy (Coleman et al., 2019). These metrics often leverage training dynamics, as they provide insight into how certain data points are learned or forgotten during training, reflecting the inherent difficulty of the data. Unlike scores computed from a single model, training dynamics-based scores capture more information across multiple training epochs, leading to a more accurate estimation of data difficulty. A more detailed discussion of training dynamic metrics is presented in Section 3.2. Besides data difficulty metrics, recent studies have refined the difficulty score methods by proposing new sampling strategies that ensure the diversity of selected examples, rather than just selecting challenging ones (Zheng et al., 2023; Maharana et al., 2023; Xia et al., 2023).

**Label-free Coreset Selection.** While supervised coreset selection methods achieve good performance, exploration in label-free select settings is limited. Most label-free coreset selection methods fall under the category of *geometry-based methods*, which select samples by leveraging geometric properties in the embedding space. This includes distances to centroids (Xia et al., 2023), to other examples (Sener & Savarese, 2017), or to the decision boundary (Ducoffe & Precioso, 2018; Margatina et al., 2021). Sorscher et al. (2022a) propose to compute a self-supervised pruning metric by performing k-means clustering in the embedding space of SWaV (Caron et al., 2020). Similarly, Xie et al. (2024) selects data via distance-based sampling of semantic patterns from the intermediate model feature. Maharana et al. (2023) introduce a method that starts with a uniform data difficulty score, which is then refined through forward and reverse messages passing on the dataset graph. Despite those efforts, label-free coreset selection still has a large performance gap compared to supervised coreset selection methods (See Figure 3). A potential reason for this gap is that SOTA label-free coreset selection methods are unable to utilize training dynamics to calculate difficulty scores, as ground truth labels are required to train models and collect these dynamics. This makes the calculation of high-quality difficulty scores a key challenge in label-free coreset selection.

## 2.2 DEEP CLUSTERING

Deep clustering aims to group unlabeled data into clusters based on learned features and has made significant strides with contrastive learning techniques (Chen et al., 2020; He et al., 2020; Van Gansbeke et al., 2020; Grill et al., 2020). Recent methods like TEMI (Adaloglou et al., 2023) and SIC (Cai et al., 2023) leverage self-supervised Vision Transformers (ViTs) and vision-language models, such as CLIP (Radford et al., 2021), DINO (Caron et al., 2021), and DINO v2 (Oquab et al., 2023). These approaches have set new deep clustering baselines on datasets such as CIFAR10/100 and ImageNet, showcasing the efficacy of leveraging rich, pretrained representations from self-supervised and multi-modal models for the deep clustering task.

## 2.3 ACTIVE LEARNING

Active learning optimizes model accuracy with minimal labeled data by iteratively requesting labels from an oracle (e.g. a human annotator) for the most informative samples (Killamsetty et al., 2021b; Ash et al., 2019; Hacohen et al., 2022). While it shares some similarities with our one-shot label-free coreset selection setting, several key differences distinguish the two directions: (1) **Selection strategies**. One-shot label-free coreset selection aims to select the coreset for human annotation in a single pass before training. However, active learning requires repeated annotation during the training process. For instance, BADGE (Ash et al., 2019) needs annotation after each training iteration. This iterative annotation setting makes it less practical as it needs continuous human involvement for each iteration. (2) **Model independence**: One-shot coreset selection aims to find a small, model-agnostic subset for training new models from scratch. In contrast, active learning selects instances tailored to the current model's needs, making it inherently model-dependent (Attenberg & Provost, 2011; Jelenić et al., 2023). Those two differences make label-free one-shot coreset selection a more practical technique for data efficiency compared to active learning. However, for a more comprehensive comparison, we include BADGE (Ash et al., 2019), a pool-based active learning method, as a label-free coreset selection baseline in our evaluation.

## 3 PRELIMINARY

In this section, we provide a brief introduction to label-free coreset selection problem definition, commonly used data difficulty scores, and sampling methods for coreset selection.

## 3.1 PROBLEM FORMULATION

The goal of label-free coreset selection is to minimize the human annotation cost by selecting an informative and representative subset while achieving good model performance. Given an unlabeled dataset $\mathcal{D} = \{x_1, x_2, ..., x_N\}$ belonging to $C$ classes and a budget $k \leq N$, we want to choose a subset $\mathcal{S} \subset \mathcal{D}$ and $|S| = k$ for labeling that maximizes the test accuracy of models trained on this labeled subset. This unsupervised one-shot coreset selection problem can be formulated as the following optimization problem:

$$S^* = \underset{\mathcal{S} \subset \mathcal{D} : |S| = k}{\arg\min} \mathbb{E}_{x,y \sim P}[l(x, y; h_\mathcal{S})], \tag{1}$$

where $P$ is the underlying distribution of $\mathcal{D}$, $l$ is the loss function, and $h_\mathcal{S}$ is the model trained with labeled $\mathcal{S}$. We assume the knowledge of the number of classes $C$, as it's needed to create an annotation scheme for human annotators and is generally assumed by other label-free data selection methods (Maharana et al., 2023; Sorscher et al., 2022a).

## 3.2 TRAINING DYNAMIC-BASED DATA DIFFICULTY SCORES

The data difficulty score of an example measures how difficult a model learns this example. For instance, the forgetting score (Toneva et al., 2018) quantifies how often a sample, once correctly classified, is subsequently misclassified during training. EL2N (Paul et al., 2021) quantifies data difficulty by calculating L2 training loss of an example in the first several epochs. Besides, the area under the margin (AUM) (Pleiss et al., 2020) measures data difficulty by accumulating margin across different training epochs. The margin for example $(\mathbf{x}, y)$ at training epoch $t$ is defined as: $M^{(t)}(\mathbf{x}, y) = z_y^{(t)}(\mathbf{x}) - \max_{i \neq y} z_i^{(t)}(\mathbf{x})$, where $z_i^{(t)}(\mathbf{x})$ is the prediction likelihood for class $i$ at training epoch $t$. AUM is the accumulated margin across all training epochs: $\mathbf{AUM}(\mathbf{x}, y) = \frac{1}{T} \sum_{t=1}^{T} M^{(t)}(\mathbf{x}, y)$. However, all those metrics are computed by training a model with those data, which requires ground truth labels.

## 3.3 CORESET SAMPLING METHODS

Early coreset selection works (Toneva et al., 2018; Paul et al., 2021; Coleman et al., 2019) advocated selecting hard examples to form coresets, which is inspired by SVM: hard examples are closer to the decision boundary, thus playing a more important role in training to determine the decision boundary. However, recent studies (Zheng et al., 2023; Maharana et al., 2023) find that data coverage is also a crucial factor for coreset selection. For instance, CCS (Zheng et al., 2023) introduces a method that combines hard example pruning with stratified sampling to jointly consider data difficulty and coverage, significantly improving coreset selection performance. Similarly, D2 (Maharana et al., 2023) refines difficulty scores by considering the difficulty of neighboring examples within a dataset graph, allowing the selection of coresets that capture data difficulty and diversity.

# 4 METHODOLOGY

In this section, we first formulate the label-free coreset problem studied in this paper. Then, we present a novel coreset selection algorithm, *ELFS* (Effective Label-Free Coreset Selection), aiming to select high-quality coresets without ground truth labels.

## 4.1 METHOD OVERVIEW

As discussed in Section 2.1, the primary limitation of existing label-free data difficulty scores lies in their failure to incorporate training dynamics, which better capture the model's learning behavior and provide a more accurate estimation of data difficulty compared to geometry-based metrics. However, existing methods for calculating training dynamic scores require ground truth labels, as these scores are derived from training models with labeled data.

To address two primary challenges discussed in Section 1, we propose *ELFS*: **1)** We propose using pseudo-labels to approximate training dynamic scores, and we employ deep clustering to generate pseudo-labels (Adaloglou et al., 2023) (Section 4.3). **2)** We find that simply using existing sampling algorithms tends to choose too many easy examples, which leads to inferior performance. We propose a simple but effective double-end pruning algorithm to mitigate this bias. The pipeline of ELFS is illustrated in Figure 2.

## 4.2 LABEL-FREE TRAINING DYNAMICS BASED SCORES ESTIMATION

Deep clustering is a technique that combines deep learning with clustering methods to group data into clusters without explicit supervision. The goal of deep clustering is to learn representations

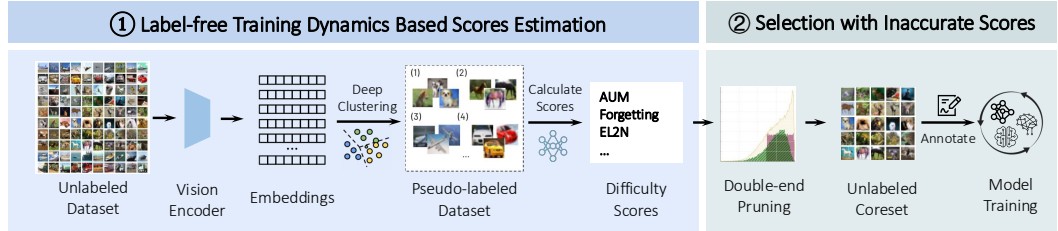

Figure 2: ***ELFS* pipeline**: (1) *Training dynamics score estimation*: *ELFS* begins with calculating image embeddings and nearest neighbors using a vision encoder and then assigns pseudo-labels to unlabeled data via deep clustering algorithms. The pseudo-labeled dataset is then used to compute training dynamics scores. (2) *Coreset selection with proxy difficulty scores*: With the pseudo-label-based scores, *ELFS* performs double-end pruning to select the unlabeled coreset. Subsequently, Humans annotate the selected coreset. This labeled coreset is used for later training.

that capture the underlying structure of the data, which are then used to assign meaningful cluster labels. In our approach, we utilize Teacher Ensemble-weighted pointwise Mutual Information (TEMI) (Adaloglou et al., 2023), a state-of-the-art deep clustering method, to generate pseudo-labels for the unlabeled dataset. We also conduct an ablation study of using different clustering methods e.g. k-means and SCAN (Caron et al., 2020) in Section 5.3.1.

TEMI uses vision encoder models (like SwAV (Caron et al., 2020) and DINO (Caron et al., 2021)) to convert all images into embedding space. Then, for each example, $x$, TEMI calculates the $k$ nearest neighbor set $\mathcal{N}_x$. After getting the nearest neighbor sets for all examples, TEMI trains an ensemble of classification heads to assign pseudo labels: TEMI uses an ensemble of student head $h_s(\cdot)$ and a teacher head $h_t(\cdot)$ that share the same architecture but differ in their updates. Each instance $x$ is processed through both types of heads, producing probabilistic classifications $q_s(c|x)$ and $q_t(c|x)$, respectively, where $c \in \{1, \ldots, C\}$ denotes the class label. The student head's parameters $\theta_s$ are updated via backpropagation, while the teacher head's parameters $\theta_t$ are updated via an exponential moving average (EMA) of the student's parameters, ensuring more stable target distributions.

**Loss function.** Given $H$ clustering heads, TEMI aims to minimize the following accumulated weighted pointwise mutual information (PMI) objective to encourage the model to align instances that are likely to belong to the same cluster while penalizing those that are less likely to be related:

$$\mathcal{L}_{\text{TEMI}}(x) = -\frac{1}{2H} \sum_{h=1}^{H} \sum_{x' \in \mathcal{N}_x} w_h(x, x') \cdot \left( \text{pmi}^h(x, x') + \text{pmi}^h(x', x) \right) \qquad (1)$$

where $\text{pmi}(x, x') = \log \left( \sum_{c=1}^{C} \frac{(q_s(c|x)q_t(c|x'))^\beta}{q_t(c)} \right)$ [2] approximates the pointwise mutual information between pairs $(x, x')$, and $w(x, x') = \sum_{c=1}^{C} q_t(c|x)q_t(c|x')$ weights each pair to reflect the probability that they belong to the same cluster.

With trained classification heads, TEMI assigns the pseudo label $\tilde{y}$ by aggregating these predictions and selecting the class with the highest combined probability:

$$\tilde{y} = \arg \max_{c \in \{1, \ldots, C\}} \frac{1}{H} \sum_{h=1}^{H} q_h(c|x)$$

Pseudo-labels generated by deep clustering make it feasible to calculate training dynamics scores with supervised training. However, the clustering-based pseudo-labeling can introduce many label noises (Table 5), which causes the distribution shift of difficulty scores. In the next part, we discuss how we address this distribution shift issue with a simple but effective double-pruning algorithm.

---

[2] $\beta$ is a hyperparameter for positive pair alignment, we follow TEMI to set it as 0.6 to avoid degeneration.

### 4.3 Coreset Selection with Proxy Difficulty Scores

#### 4.3.1 Observation: Importance Score Distribution Shift

Given the difficulty scores calculated with pseudo labels, a straightforward approach is to directly apply existing coreset selection methods, such as CCS (Zheng et al., 2023), to select coreset with pseudo-label-based scores. Surprisingly, despite pseudo-labels are not always correct, simply applying CCS on pseudo-label-based score (AUM) outperforms SOTA label-free coreset selection methods (green curve in Table 3). However, there is still a performance gap between CCS with pseudo-label scores (green curve) and CCS with ground truth label scores (purple dashed curve).

**Biased coverage of CCS with pseudo-labels data difficulty scores.** To better understand the reason behind this gap, we compared the ground truth AUM distribution of coresets selected by CCS (supervised) and CCS (pseudo-label). In Figure 4(a), we plot the distribution of AUM calculated with the ground truth labels and the distribution of examples selected by CCS on this ground truth distribution. To make the comparison, we plot the distribution of ground truth AUM for the examples chosen by CCS using pseudo-label-based AUM. From the figure, we can observe a significant distribution shift between CCS (supervised) and CCS (pseudo-label). Specifically, pseudo-label-based CCS (striped green region) selects more easy examples (high-AUM) compared to CCS using ground truth labels (purple region), which can be a potential reason leading to the performance drop.

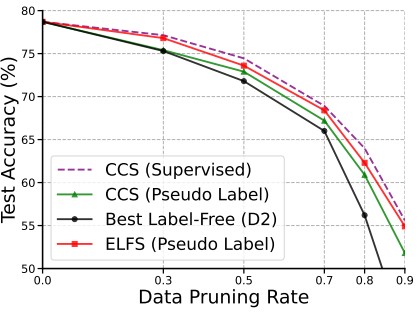

Figure 3: Performance comparison of supervised CCS (Zheng et al., 2023), label-free CCS, best label-free baseline (label-free D2 (Maharana et al., 2023)), and our method *ELFS* on CIFAR100.

#### 4.3.2 Pruning with Pseudo-Label-Based Scores

To address this distribution gap of selected coresets, we propose a simple but effective double-end pruning method to select data with pseudo-label-based scores, which aims to reduce the number of selected easy examples. Double-end pruning consists of two steps: **1)** Prune $\beta$ hard examples first, where $\beta$ is a hyperparameter (which is also used in (Maharana et al., 2023; Zheng et al., 2023; Mindermann et al., 2022)). **2)** Continue pruning easy examples until the budget is met.

A key challenge in our double-end pruning algorithm is determining the hard pruning rate $\beta$. Recent SOTA coreset selection works (Zheng et al., 2023; Maharana et al., 2023) show that hard example pruning plays a crucial role in improving the coreset selection performance, since it helps ensure the selected coresets better cover high-density areas (Zheng et al., 2023). A common practice is to perform a grid search to select the optimal $\beta$. In supervised settings, methods like CCS (Zheng et al., 2023) and D2 (Maharana et al., 2023) can grid search the best $\beta$ with ground truth labels. However, in label-free coreset selection, the absence of ground truth labels makes the $\beta$ grid search a challenge.

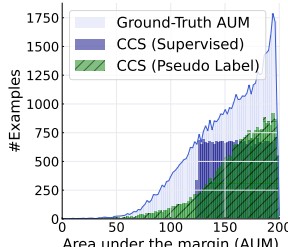
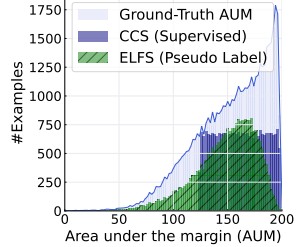

(a) Label-free CCS coreset.  (b) *ELFS* coreset.

Figure 4: Ground truth label AUM distribution of different coreset on CIFAR100. The pruning rate for coreset is 50%. After applying double-end (*ELFS*) pruning on data with pseudo-label-based scores, *ELFS* covers more hard data in the selected coreset when mapped onto ground truth AUM.

To address this challenge, we show that pseudo-labels generated in the deep clustering step provide a good estimation for this $\beta$ search: After generating the pseudo-labeled dataset, we split it into 90% for training and 10% for validation. We use the validation set to determine the optimal $\beta$. Once the optimal $\beta$ is identified, *ELFS* selects coresets from the entire training dataset. Our evaluation shows that $\beta$ grid search with pseudo-labels provides a good estimation for the optimal $\beta$. With this

Table 1: Comparison of pretrained vision models' performance. ELFS consistently outperforms baselines across all encoders. Full dataset training results: CIFAR10 - 95.5%, CIFAR100 - 78.7%, ImageNet-1K - 73.1%. Badge represents the active learning (AL) baseline, while the remaining baselines are based on coreset selection. Red-highlighted cells indicate performance below random sampling. The top-performing result for each encoder is bolded.

| Encoder | Pruning Rate | CIFAR10 | | | | | CIFAR100 | | | | | ImageNet-1K | | | | |
|---|---|---|---|---|---|---|---|---|---|---|---|---|---|---|---|---|
| | | 30% | 50% | 70% | 80% | 90% | 30% | 50% | 70% | 80% | 90% | 30% | 50% | 70% | 80% | 90% |
| - | Best Supervised | 95.7 | 94.9 | 93.3 | 91.4 | 87.1 | 78.2 | 75.9 | 70.5 | 65.2 | 56.9 | 72.9 | 71.8 | 68.1 | 65.9 | 55.6 |
| - | Random | 94.3 | 93.4 | 90.9 | 88.0 | 79.0 | 74.6 | 71.1 | 65.3 | 57.4 | 44.8 | 72.2 | 70.3 | 66.7 | 62.5 | 52.3 |
| | Badge (AL) | 93.6 | 93.0 | 91.0 | 87.9 | 81.6 | 74.7 | 71.8 | 65.2 | 58.9 | 47.8 | 71.7 | 70.4 | 65.8 | 61.7 | 53.4 |
| SwAV | Prototypicality | 94.7 | 92.9 | 90.1 | 84.2 | 70.9 | 74.5 | 69.8 | 61.1 | 48.3 | 32.1 | 70.9 | 60.8 | 54.6 | 41.9 | 30.6 |
| | D2 | 94.3 | 93.8 | 91.6 | 85.1 | 71.4 | 75.3 | 71.3 | **66.0** | 56.2 | 42.1 | 72.3 | 65.6 | 55.6 | 50.8 | 43.2 |
| | ELFS (Ours) | **95.0** | **94.3** | **91.8** | **89.8** | **82.5** | **76.1** | **72.1** | 65.5 | **58.2** | **49.8** | **73.2** | **71.4** | **66.8** | **62.7** | **53.4** |
| DINO | FreeSel | 94.5 | 93.8 | 91.7 | 88.9 | 82.4 | 75.0 | 70.5 | 65.6 | 57.6 | 44.8 | 72.2 | 70.0 | 65.4 | 61.0 | 51.1 |
| | ELFS (Ours) | **95.5** | **95.2** | **93.2** | **90.7** | **87.3** | **76.8** | **73.6** | **68.4** | **62.3** | **54.9** | **73.5** | **71.8** | **67.2** | **63.4** | **54.9** |

pseudo-label-based $\beta$ search, ELFS guarantees that **ELFS does not use any ground truth labels in the coreset selection process**. We conduct a more detailed analysis on $\beta$ search in Section 5.3.6.

Despite its simplicity, we find that our double-end pruning scheme significantly reduces the number of selected easy examples (striped green region in Figure 4(b)) and improves the performance of label-free coreset selection (purple curve in Table 3).

# 5 EXPERIMENTS

In this section, we evaluate *ELFS* on four vision benchmarks. Our evaluation results show that *ELFS* outperforms existing label-free coreset selection baselines at all pruning rates. When using SwAV as the encoder, *ELFS* outperforms D2 by up to 10.2% in accuracy on ImageNet. Similarly, when using DINO as the encoder, *ELFS* outperforms FreeSel with an accuracy gain of up to 3.9% on ImageNet.

## 5.1 EXPERIMENT SETTING

**Label-free coreset baselines.** We compare *ELFS* with the following label-free coreset selection baselines: 1) **Random**: Randomly sampled coreset. 2) **Prototypicality** (Sorscher et al., 2022b): Prototypicality generates embeddings from a pretrained vision model (SwAV (Caron et al., 2020)) to perform k-means clustering. The difficulty of each data point is quantified by its Euclidean distance to the nearest cluster centroid, with those further away being preferred during coreset selection. 3) **Label-free D2** (Maharana et al., 2023): D2 adapts supervised D2 pruning, which uses difficulty scores derived from training dynamics and selects coresets through forward and backward message passing in a dataset graph. In a label-free scenario, a uniform initial difficulty score is applied, and the dataset graph is initialized with embeddings from SwAV (Caron et al., 2020). 4) **FreeSel** (Xie et al., 2024): distance-based sampling on patterns extracted from inter-mediate features of pretrained models. 5) **BADGE** (Ash et al., 2019): An active learning method that selects new samples in each epoch using k-means++ initialization within the gradient vector space. It is important to note that BADGE itself is not a coreset selection method, and we include it as a reference for comparison with active learning approaches.

**Implementation.** We report *ELFS* results using DINO (Caron et al., 2021) and SwAV (Caron et al., 2020) as the pretrained vision model to extract embeddings. We use the area under the margin (AUM) (Pleiss et al., 2020) as the training dynamics metric for all datasets. Our ablation study in Section 5.3.3 show that the choice of training dynamic metrics only has a marginal impact on *ELFS*'s performance. We use ResNet-34 (He et al., 2016) for ImageNet-1K, and ResNet-18 (He et al., 2016) for the rest. Due to the space limitation, we include more experimental details in Appendix B. All reported results are based on training from scratch using the selected coreset.

## 5.2 LABEL-FREE CORESET SELECTION PERFORMANCE COMPARISON

To verify the effectiveness of ELFS, we evaluate and compare ELFS with other baselines on four vision benchmarks. We report the comparison results on CIFAR10, CIFAR100, and ImageNet-1K in Table 1. (STL10 results can be found in Table 10 in the Appendix). As discussed in Section 2, vision encoders are widely adopted in label-free coreset selection methods. To guarantee a fair comparison, we report the performance of *ELFS* with the same encoders used by other baselines in their papers.

As shown in Table 1, given the same vision encoder, *ELFS* consistently outperforms existing label-free coreset selection baselines. For instance, when using SwAV as the encoder, *ELFS* outperforms D2 by up to 10.2% in accuracy on ImageNet-1K. Similarly, when employing DINO as the encoder, *ELFS* outperforms FreeSel with an accuracy gain of up to 3.9% on ImageNet-1K. Moreover, for some pruning rates (e.g., 30% and 50% for ImageNet-1K), *ELFS* even achieves comparable performance compared to the best supervised coreset selection performance. We highlight in red the cells where the selected coreset performs worse than random sampling. Notably, random sampling remains a strong baseline, particularly at higher pruning rates (80% to 90%). However, *ELFS* consistently outperforms both random sampling and other existing baselines. To the best of our knowledge, *ELFS* is the *first label-free approach* to outperform random sampling at all pruning rates. Even though active learning and label-free one-shot coreset selection are two different types of methods (as discussed in Section 2.3). For a more comprehensive comparison, we also compare our method with an active learning baseline, Badge (Ash et al., 2019). For STL10, we observe similar findings—*ELFS* outperforms all existing label-free baselines (Table 8 in Appendix C.).

## 5.3 ABLATION STUDY & ANALYSIS

### 5.3.1 ABLATION STUDY: CLUSTERING METHOD

Table 2: Ablation study on different pseudo-label generation methods—(1) deep clustering: TEMI and SCAN, and (2) K-Means. All clustering methods use the DINO embeddings.

| Pseudo-Label ACC | | | CIFAR10 | | | | | CIFAR100 | | | | |
|---|---|---|---|---|---|---|---|---|---|---|---|---|
| CIFAR10 | CIFAR100 | Pruning Rate | 30% | 50% | 70% | 80% | 90% | 30% | 50% | 70% | 80% | 90% |
| 92.5% | 66.3% | ELFS (DINO + TEMI) | **95.5** | **95.2** | 93.2 | 90.7 | **87.3** | **76.8** | **73.6** | **68.4** | **62.3** | **54.9** |
| 93.7% | 60.6% | ELFS (DINO + SCAN) | 95.3 | **95.2** | **94.0** | **90.8** | **87.3** | 75.9 | 72.5 | 67.8 | 61.6 | 53.7 |
| 88.3% | 57.7% | ELFS (DINO + K-Means) | 95.3 | 94.1 | 93.3 | 90.3 | 85.5 | 75.2 | 73.2 | 67.4 | 54.9 | 29.0 |
| | - | Best Label-Free | 94.7 | 93.8 | 91.7 | 88.9 | 82.4 | 75.3 | 71.8 | 66.0 | 58.9 | 44.8 |
| | - | Random | 94.3 | 93.4 | 90.9 | 88.0 | 79.0 | 74.6 | 71.1 | 65.3 | 57.4 | 44.8 |

To understand the impact of different clustering methods on *ELFS*, we select two deep clustering methods—SCAN (Van Gansbeke et al., 2020) and TEMI (Adaloglou et al., 2023)—along with K-Means to evaluate their influence on the performance of *ELFS*. As shown in Table 2, the results reveal that as the quality of pseudo-labels decreases, the performance of the selected coreset also degrades. Both TEMI and SCAN show strong performance as the clustering methods for *ELFS*.

### 5.3.2 ABLATION STUDY: SAMPLING METHOD

Table 3: Ablation study on different pseudo-label training dynamics sampling methods on CI-FAR100. The coresets are selected using three sampling methods—CCS, D2, and ELFS (double-end pruning)—on the AUM score derived from pseudo-label training dynamics.

| Pruning Rate | 30% | 50% | 70% | 80% | 90% |
|---|---|---|---|---|---|
| CCS (Pseudo-Label) | 75.5 | 72.9 | 67.2 | 60.9 | 51.8 |
| D2 (Pseudo-Label) | 74.8 | 70.9 | 61.2 | 49.6 | 27.7 |
| ELFS (Pseudo-Label) | **76.8** | **73.6** | **68.4** | **62.3** | **54.9** |
| CCS (Ground-Truth Label) | 77.1 | 74.5 | 68.9 | 64.0 | 56.0 |

As we discussed in Section 4.3, directly applying CCS sampling leads to a performance drop due to the score distribution shift caused by inaccurate pseudo-labels. In this section, we conduct an in-depth ablation study to compare the performance of different sampling methods on CIFAR100 with pseudo-labels generated by DINO-TEMI. As shown in Table 3, double-end pruning used in *ELFS* consistently achieves comparable and better performance than other sampling methods.

### 5.3.3 ABLATION STUDY: DATA DIFFICULTY METRIC

Table 4: Ablation study on the data difficulty scores. We report results using different training dynamic metrics—AUM, forgetting, and EL2N.

|  | CIFAR10 | | | | | CIFAR100 | | | | |
| --- | --- | --- | --- | --- | --- | --- | --- | --- | --- | --- |
| Pruning Rate | 30% | 50% | 70% | 80% | 90% | 30% | 50% | 70% | 80% | 90% |
| ELFS (AUM) | 95.5 | **95.2** | 93.2 | 90.7 | **87.3** | **76.8** | 73.6 | **68.4** | **62.3** | **54.9** |
| ELFS (Forgetting) | 95.4 | 94.9 | 93.3 | 90.1 | 86.9 | **76.8** | **74.0** | **68.4** | 61.9 | **54.9** |
| ELFS (EL2N) | **95.6** | **95.2** | **93.7** | **91.0** | 86.5 | 76.5 | 73.6 | 68.2 | 62.0 | 52.8 |

In Section 5.2, we use AUM (Pleiss et al., 2020) for ELFS. To understand how other data difficulty scores influence the performance of *ELFS*, we conduct an ablation study of the data difficulty score metrics on CIFAR10 and CIFAR100. Table 4 shows that *ELFS* with different data difficulty metrics achieves very similar coreset selection performance, which indicates that using a different score metric, like forgetting (Toneva et al., 2018) and EL2N (Paul et al., 2021), only has a marginal impact on *ELFS* performance.

### 5.3.4 PSEUDO-LABEL QUALITY

Table 5: Pseudo-labeling statistics. Reported metrics include accuracy (ACC), adjusted random index (ARI), and normalized mutual information (NMI) in %.

|  | CIFAR10 | | | CIFAR100 | | | ImageNet-1K | | |
| --- | --- | --- | --- | --- | --- | --- | --- | --- | --- |
|  | ACC (%) | NMI (%) | ARI (%) | ACC (%) | NMI (%) | ARI (%) | ACC (%) | NMI (%) | ARI (%) |
| TEMI (SwAV) | 60.7 | 54.6 | 43.4 | 39.8 | 57.7 | 26.5 | 43.1 | 73.5 | 29.5 |
| TEMI (DINO) | 92.5 | 86.5 | 85.1 | 66.3 | 76.5 | 53.0 | 58.8 | 81.4 | 44.8 |

Pseudo-labels generated by deep clustering are not guaranteed to be correct. In Table 5, we report the accuracy of pseudo-labels on CIFAR10, CIFAR100, and ImageNet-1K. As shown in the table, pseudo-labels contain a considerable amount of label noise. For ImageNet-1K, accuracy is below 60% even using DINO as the vision encoder. This indicates that *ELFS* has strong robustness to label noise. Despite the label noise, *ELFS* still achieves better performance than all baselines.

### 5.3.5 VISION ENCODER TRAINED FROM SCRATCH ON THE SAME UNLABELLED DATASET

Table 6: Ablation study results comparing model performance when public pretrained models are unavailable. We use a self-supervised learning method (Caron et al., 2020) to train vision encoders from scratch with all unlabeled data.

|  | CIFAR10 | | | | | ImageNet-1K | | | | |
| --- | --- | --- | --- | --- | --- | --- | --- | --- | --- |
| Pruning Rate | 30% | 50% | 70% | 80% | 90% | 30% | 50% | 70% | 80% | 90% |
| ELFS (DINO) | 95.5 | 95.2 | 93.2 | 90.7 | 87.3 | 73.5 | 71.8 | 67.2 | 63.4 | 54.9 |
| Random | 94.3 | 93.4 | 90.9 | 88.0 | 79.0 | 72.2 | 70.3 | 66.7 | 62.5 | 52.3 |
| FreeSel (DINO) | 94.5 | 93.8 | 91.7 | **88.9** | 82.4 | 72.2 | 70.0 | 65.4 | 61.0 | 51.1 |
| D2 (SwAV) | 94.3 | 93.8 | 91.6 | 85.1 | 71.4 | 72.3 | 65.6 | 55.6 | 50.8 | 43.2 |
| ELFS (Self-Trained Encoder) | **94.8** | **94.6** | **92.4** | **88.9** | **82.8** | **73.2** | **71.4** | **66.8** | **62.7** | **53.4** |

In our previous evaluation, we adopt public pretrained encoders to extract embeddings from images (to make a fair comparison with existing baselines). In this section, we study how ELFS performs without a pretrained encoder: we use a self-supervised learning method (Caron et al., 2020) to train an encoder with all unlabeled data and report results in Table 6. Our evaluation results show that, despite that *ELFS* performance declines after replacing DINO with the self-trained encoders, *ELFS* with the self-trained encoder still maintains strong performance and outperforms other label-free coreset selection baselines using a pretrained encoder. We argue that with the increasing availability of publicly released pretrained models, high-quality vision encoders have become more accessible. However, our study suggests that even without a public pretrained encoder, self-supervised learning on the same unlabeled dataset can be an alternative to obtaining a vision encoder.

### 5.3.6 Hard Pruning Rate $\beta$ Search with Pseudo-Labels

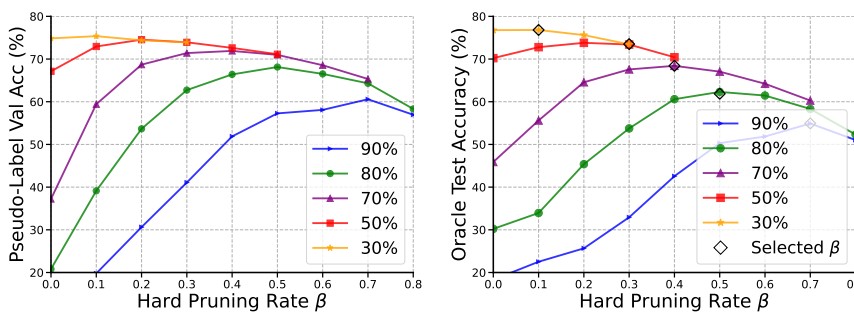

(a) DINO pseudo-labal validation accuracy.    (b) DINO ground truth label test accuracy.

Figure 5: (a) reports the pseudo-label validation accuracy on different $\beta$ on CIFAR100. (b) reports the ground truth (oracle) label test accuracy on different $\beta$. Curves with different colors stand for different pruning rates. $\diamond$ indicates the $\beta$ selected by the best pseudo-label validation accuracy, which is equal to or close to the optimal $\beta$.

As discussed in Section 4.3, *ELFS* has no access to ground truth labels for searching the optimal $\beta$. To address this challenge, we propose using pseudo-labels to choose $\beta$. In this section, we conduct an in-depth study on the quality of $\beta$ selected by pseudo-labels. Figure 5 compares the pseudo-label accuracy (a) and ground truth accuracy (b) under different hard pruning rates $\beta$ with a $0.1$ search step. $\diamond$ in Figure 5(b) indicates the $\beta$ selected by the best pseudo-label validation accuracy. We find that $\beta$ selected by pseudo-labels provides a good estimation for the optimal $\beta$ selected by ground truth labels. Our results on CIFAR100 show that, when pruning rate is 30%, 70%, 80%, and 90%, pseudo-label search selects the same $\beta$ as the ground truth label search. At a $50\%$ pruning rate, the accuracy of the model trained with the coreset chosen with pseudo-labels is only $0.1\%$ lower than the coreset chosen with ground truth labels.

### 5.3.7 Transferability

Table 7: Performance of different architectures on CIFAR10 at varying pruning rates, trained with coreset selection based on forgetting scores from ResNet18 DINO pseudo-label training dynamics. We present the top-performing results from various baselines as the best SOTA performance.

| Pruning Rate | 30% | 50% | 70% | 80% | 90% |
|---|---|---|---|---|---|
| Random (ResNet18) | 94.3 | 93.4 | 90.9 | 88.0 | 79.0 |
| Best SOTA Label-Free | 94.7 | 93.8 | 91.7 | 88.9 | 82.4 |
| *ELFS* (ResNet18) | **95.5** | 95.2 | **93.2** | 90.7 | 87.3 |
| *ELFS* (ResNet34) | 95.4 | **95.3** | 93.1 | **90.9** | **87.6** |
| *ELFS* (ResNet50) | 95.4 | 94.8 | 93.1 | 90.3 | 87.3 |

One application of coreset selection is that a selected coreset can also be used for training other models. In this section, we evaluate coreset transferability on CIFAR10 using coresets selected during ResNet18 training. As shown in Table 7, these coresets exhibit good transferability to ResNet34 and ResNet50, demonstrating robustness and generalizability across different architectures.

## 6 Conclusion

In this paper, we present a novel label-free coreset selection method, *ELFS*, to select high-quality coresets without ground truth labels. *ELFS* first estimates data difficulty scores with pseudo-labels generated by deep clustering and then performs double-end pruning to select a coreset with these inaccurate data difficulty scores. By evaluating *ELFS* on four vision benchmarks, we show that *ELFS* consistently outperforms SOTA label-free coreset selection baselines and, in some cases, nearly matches the performance of supervised coreset selection methods, which demonstrates the effectiveness of *ELFS*. Label-free coreset selection can significantly reduce data collection costs. We believe that our work provides a strong baseline and will inspire future work in this area.

ACKNOWLEDGEMENT

This work was partially supported by Cisco Research Grant. Any opinions, findings, and conclusions or recommendations expressed in this material are those of the author(s) and do not necessarily reflect the views of our research sponsors.

Prepared by LLNL under Contract DE-AC52-07NA27344 and supported by the LLNL-LDRD Program under Project No. 24-ERD-010 and Project No. 23-ERD-030 (LLNL-CONF-2000176). This manuscript has been authored by Lawrence Livermore National Security, LLC under Contract No. DE-AC52-07NA27344 with the U.S. Department of Energy. The United States Government retains, and the publisher, by accepting the article for publication, acknowledges that the United States Government retains a non-exclusive, paid-up, irrevocable, world-wide license to publish or reproduce the published form of this manuscript, or allow others to do so, for United States Government purposes.

REPRODUCIBILITY

We include implementation details in Section 5.1 and Section B in the Appendix. The implementation of *ELFS* is available at https://github.com/eltsai/elfs.

ETHICS STATEMENT

Our work on coreset selection using pseudo-labels and double-end pruning aligns with the ICLR Code of Ethics. By reducing reliance on labeled data, our method lowers annotation costs, enhances scalability, and promotes sustainable AI by improving training efficiency and reducing the carbon footprint of large-scale models. We also recognize the broader societal impact of this work, particularly in democratizing access to state-of-the-art machine learning techniques. By reducing the need for extensive labeled datasets, our approach can be adopted by researchers in resource-constrained environments.

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

## A  OVERVIEW

In Section B, we provide details on datasets, best hyperparameters for our models, and other experiment settings.

In Section C, we report additional evaluation results.

**Datasets**. CIFAR10, CIFAR100 (Krizhevsky et al., 2009): Custom Licence. ImageNet-1K (Deng et al., 2009): Custom Licence. STL10 (Coates et al., 2011): Custom Licence.

## B  DETAILED EXPERIMENT SETTING

### B.1  PSEUDO-LABEL GENERATION

For pseudo-label generation, we use the training settings recommended in TEMI (Adaloglou et al., 2023). For CIFAR10, CIFAR100 (Krizhevsky et al., 2009) and STL10 (Coates et al., 2011), we calculate the 50-nearest neighbors (50-NN) for each example using cosine distance in the pretrained model's embedding space. For clustering, 50 heads are trained. Training is conducted over 200 epochs with a batch size of 512, using an AdamW optimizer (Loshchilov & Hutter, 2017b) with a learning rate of 0.0001 and a weight decay of 0.0001. For ImageNet (Deng et al., 2009), all the parameters are the same, except the nearest neighbor search is adjusted to 25-nearest neighbors (25-NN).

### B.2  CORESET SELECTION WITH PSEUDO-LABEL

Table 8: Comparison of pretrained vision models' performance on STL10. ELFS consistently outperforms baselines across all encoders. Red-highlighted cells indicate performance below random sampling. The top-performing result for each encoder is bolded.

| Encoder | Pruning Rate | STL10 30% | 50% | 70% | 80% | 90% |
|---|---|---|---|---|---|---|
| - | Random | 75.6 | 70.2 | 64.0 | 57.9 | 45.4 |
|  | Badge (AL) | 68.4 | 63.7 | 57.9 | 51.0 | 44.1 |
| SwAV | Prototypicality | 66.5 | 60.5 | 43.2 | 37.6 | 20.8 |
|  | D2 | 76.2 | 69.2 | 58.2 | 52.7 | 40.2 |
|  | ELFS (Ours) | **79.6** | **77.9** | **72.5** | **71.6** | **49.2** |
| DINO | FreeSel | 77.3 | 72.4 | 62.5 | 55.8 | 45.3 |
|  | ELFS (Ours) | **78.2** | **73.9** | **65.7** | **61.3** | **52.7** |

**Dataset Specific Training Details**. We benchmark our method *ELFS* using identical training settings as recommended in CCS (Zheng et al., 2023) and D2 (Maharana et al., 2023). **CIFAR10 and CIFAR100**: We use a ResNet18 model for 40,000 iterations, with a batch size of 256 and SGD optimizer settings that include 0.9 momentum and 0.0002 weight decay. The initial learning rate is set at 0.1 with a cosine annealing learning rate scheduler (Loshchilov & Hutter, 2017a). **ImageNet-1K**: we train a ResNet34 model for 300,000 iterations, maintaining the same optimizer parameters and initial learning rate. **STL10**: We utilize the labeled portion of the STL10 dataset for training. For all coresets with different pruning rates, we train models with 6,400 iterations with a batchsize of 256 (about 320 epochs when pruning rate is zero). We use the SGD optimizer (0.9 momentum and 0.0002 weight decay) with a 0.1 initial learning rate. Identical to other datasets, we use a cosine annealing learning rate scheduler (Loshchilov & Hutter, 2017a) with a 0.0001 minimum learning rate.

**Coreset Selection.** For pruning ratio $\alpha = \{30\%, 50\%.70\%, 80\%, 90\%\}$, we report $\beta$ for the best coreset start point selected in Table 9 on a variety of datasets.

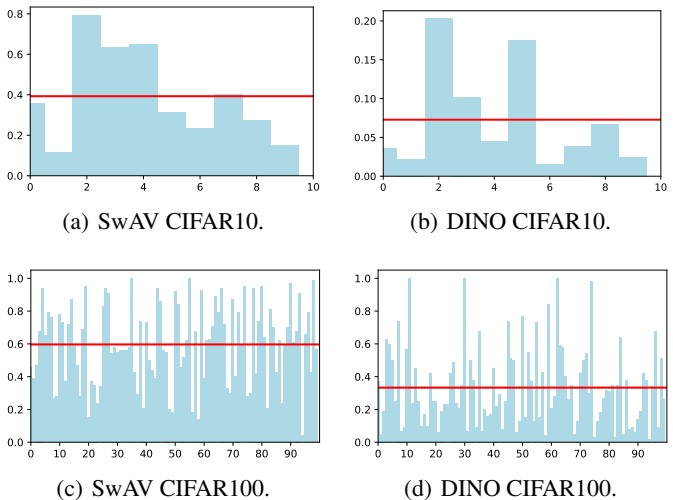

Figure 6: Class-wise misclassification rate on CIFAR10/100 using pseudo-labels from different pretrained models: SwAV and DINO. See Table 5 for the overall misclassification rate.

Table 9: Dataset specific pruning rates and their corresponding chosen $\beta$.

| Dataset | Encoder | Pruning Rate | | | | |
|---|---|---|---|---|---|---|
| | | 30% | 50% | 70% | 80% | 90% |
| CIFAR-10 | SwAV | 0.1 | 0.2 | 0.2 | 0.3 | 0.4 |
| | DINO | 0 | 0 | 0.1 | 0.1 | 0.4 |
| CIFAR-100 | SwAV | 0.1 | 0.2 | 0.4 | 0.5 | 0.6 |
| | DINO | 0.0 | 0.2 | 0.4 | 0.5 | 0.7 |
| ImageNet-1K | DINO | 0.0 | 0.1 | 0.2 | 0.4 | 0.5 |
| | SwAV | 0 | 0 | 0.3 | 0.4 | 0.6 |
| STL10 | DINO | 0.2 | 0.2 | 0.4 | 0.6 | 0.7 |
| | SwAV | 0.1 | 0.2 | 0.3 | 0.3 | 0.5 |

Table 10: The misclassification rate of pseudo-labels vs ground-truth labels, adjusted random index (ARI), and normalized mutual information (NMI) using TEMI.

| | DINO | | | SwAV | | |
|---|---|---|---|---|---|---|
| | ACC (%) | NMI (%) | ARI (%) | ACC (%) | NMI (%) | ARI (%) |
| STL10 | 93.0 | 89.6 | 86.1 | 89.6 | 82.0 | 79.3 |

# C  ADDITIONAL EVALUATION RESULTS

## C.1  STL-10 EVALUATION

In this section, we present evaluation results on STL10 (see Table 8). On STL10, our observations align with the results from CIFAR10/100 and ImageNet. Our method, *ELFS*, consistently outperforms other label-free coreset selection methods across various pruning rates. We provide the pseudo-label misclassification rate, normalized mutual information (NMI), and adjusted random index (ARI) in Table 10. Additionally, Table 6 shows the class-wise misclassification rate on CIFAR10 and CIFAR100, highlighting *ELFS*'s effectiveness with imbalanced class assignments.

## C.2 ABLATION STUDY WITH CLIP ENCODER

We conduct additional experiments using CLIP (Radford et al., 2021) as the encoder on CIFAR10 and CIFAR100, as shown in Table 11. The pseudo-label accuracy generated by CLIP (85.2% for CIFAR10 and 54.0% for CIFAR100) is better than SwAv (60.7% for CIFAR10 and 39.8% for CI-FAR100) but worse than DINO (92.5% for CIFAR10 and 66.3% for CIFAR100). These results align with the trends observed in TEMI (Adaloglou et al., 2023). Importantly, all methods achieve pseudo-label accuracy significantly better than SOTA label-free approaches, further highlighting the robustness of ELFS.

For instance, in CIFAR100, DINO achieves the highest performance across all pruning rates, maintaining robust coreset quality even as pruning increases, owing to its superior pseudo-label accuracy. CLIP, while outperforming SwAv, shows a decline in performance at higher pruning rates, emphasizing the importance of encoder quality in maintaining coreset effectiveness. These findings confirm the ablation results in Section 5.3.1 and Section 5.3.4, which demonstrate that better encoders result in higher-quality pseudo-labels and consequently improve the quality of the selected coresets.

Overall, these results validate that our method benefits from pseudo-labels generated by strong encoders like DINO and CLIP, outperforming SOTA label-free methods, and showing the importance of encoder quality in achieving optimal performance in label-free coreset selection.

Table 11: Ablation study on encoders with CLIP (Radford et al., 2021) on CIFAR10 and CIFAR100. We also report pseudo-label accuracy (PL-ACC) using different encoders.

| Encoder | CIFAR10 | | | | | | CIFAR100 | | | | | |
| | PL-ACC | 30% | 50% | 70% | 80% | 90% | PL-ACC | 30% | 50% | 70% | 80% | 90% |
| --- | --- | --- | --- | --- | --- | --- | --- | --- | --- | --- | --- | --- |
| DINO | 92.5% | 95.5 | 95.2 | 93.2 | 90.7 | 87.3 | 66.3% | 76.8 | 73.6 | 68.4 | 62.3 | 54.9 |
| CLIP | 85.2% | 95.0 | 94.5 | 92.7 | 90.1 | 84.5 | 54.0% | 75.5 | 73.1 | 66.5 | 58.4 | 50.6 |
| SwAv | 60.7% | 95.0 | 94.3 | 91.8 | 89.8 | 82.5 | 39.8% | 76.1 | 72.1 | 65.5 | 58.2 | 49.8 |
| Best Label-Free | | 94.7 | 93.8 | 91.7 | 88.9 | 82.4 | - | 75.3 | 71.8 | 66.0 | 58.9 | 44.8 |

## C.3 IMAGENET-1K TRANSFERABILITY STUDY

Table 12: Imagenet ResNet-50 Transferability

| Pruning Rate | 30% | 50% | 70% | 80% | 90% |
| --- | --- | --- | --- | --- | --- |
| ResNet-34 | 73.5 | 71.8 | 67.2 | 63.4 | 54.9 |
| ResNet-50 | 76.2 | 74.0 | 69.5 | 64.6 | 56.4 |

Table 12 presents the results of the transferability experiment on ImageNet-1K, where coresets were selected using the ResNet-34 and evaluated on both ResNet-34 and ResNet-50. The results show that ResNet-50 consistently outperforms ResNet-34 across all pruning rates, with performance differences increasing at higher pruning rates. For example, at a 70% pruning rate, ResNet-50 achieves 69.5% accuracy compared to 67.2% for ResNet-34, and at 90%, ResNet-50 maintains 56.4% accuracy compared to 54.9%. These findings highlight the robustness and generalizability of the selected coresets, as they effectively transfer across architectures.

## C.4 DATA DIFFICULTY METRIC CASE STUDY VISUALIZATION

In this section, we visualize the hard/easy data chosen by the area under the margin (AUM) metric (Pleiss et al., 2020). AUM measures data difficulty by accumulating margin across different training epochs. The margin for example $(\mathbf{x}, y)$ at training epoch $t$ is defined as: $M^{(t)}(\mathbf{x}, y) = z_y^{(t)}(\mathbf{x}) - \max_{i \neq y} z_i^{(t)}(\mathbf{x})$, where $z_i^{(t)}(\mathbf{x})$ is the prediction likelihood for class $i$ at training epoch $t$. AUM is the accumulated margin across all training epochs: $\mathbf{AUM}(\mathbf{x}, y) = \frac{1}{T} \sum_{t=1}^{T} M^{(t)}(\mathbf{x}, y)$.

Following Pleiss et al. (2020), we illustrate the distinction between hard and easy examples using AUM in Figure 7. The graphs depict logit values (model predictions before the softmax activation) over training epochs for one hard-to-learn image and one easy-to-learn image. For both examples, the green line represents the logit value for the correct label, while the red line represents the largest

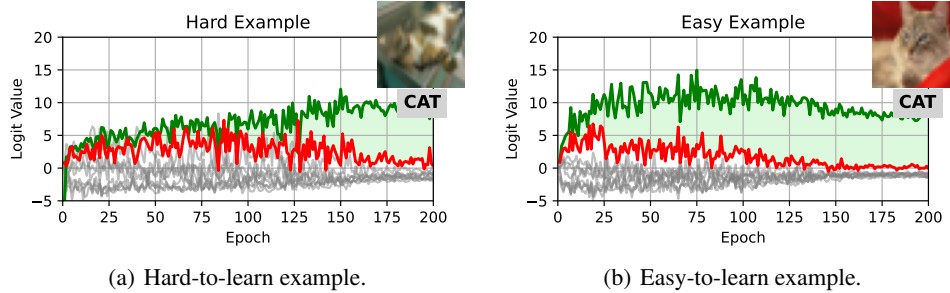

(a) Hard-to-learn example.
(b) Easy-to-learn example.

Figure 7: Illustration of the hard and easy example using the AUM metrics (Pleiss et al., 2020). The green line represents the logit for the correct label, and the red line represents the largest other logit. The green region represents positive AUM. The logits are from training a ResNet-18 Model on the CIFAR10 dataset.

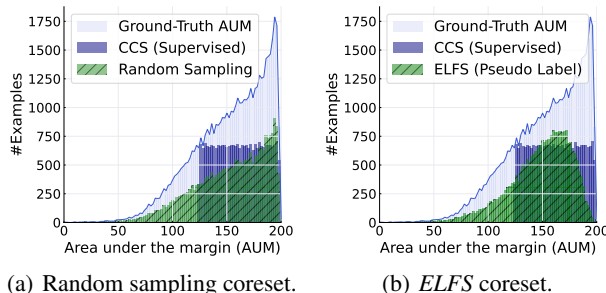

(a) Random sampling coreset.
(b) *ELFS* coreset.

Figure 8: Ground-truth label AUM distribution of different coresets on CIFAR100 with a pruning rate of 50%. Random sampling selects more easy-to-learn data due to its higher density in the dataset. In contrast, ELFS has less coverage on easy-to-learn examples. This difference explains the performance enhancement observed with ELFS (73.6%) compared to random sampling (71.1%).

logit value among incorrect labels. The gray line shows other logits. The green region indicates a positive AUM, where the correct label consistently has the highest logit value, signifying confidence in the prediction.

In the hard example (Figure 7(a)), the model struggles to confidently predict the correct label throughout training, as evidenced by fluctuations and overlap between the green and red lines. In contrast, the easy example (Figure 7(b)) shows the correct label (green line) maintaining a clear margin above the incorrect predictions (red line), indicating that the model quickly and confidently learns the example.

## C.5   DATA DISTRIBUTION COMPARISON BETWEEN ELFS AND RANDOM SAMPLING

Figure 8 compares the distribution of ground-truth label AUM across coresets selected by different methods on CIFAR100 with a 50% pruning rate. Random sampling, as shown in the green bars in Figure 8(a), tends to select more easy-to-learn examples, which are higher in density in the dataset. These examples have higher AUM values, indicating that they are easier for the model to predict. However, this results in an underrepresentation of harder-to-learn examples—data points with lower AUM values that are crucial for improving the model's robustness and generalization. In contrast, ELFS (green bars) in Figure 8(b) selects a more balanced coreset, covering more hard examples with lower AUM values by leveraging pseudo-label-based scores and double-end pruning. This targeted selection explains why ELFS achieves better performance (73.6%) compared to random sampling (71.1%). The graph highlights the importance of data difficulty diversity in the coreset to ensure a more effective and robust training process.

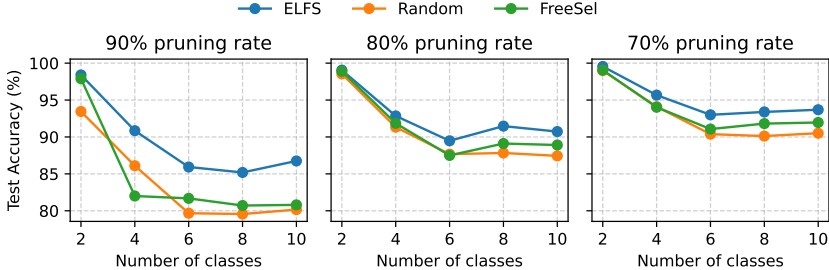

Figure 9: Coreset-based continual learning evaluation on CIFAR10. In the class incremental setting with 2 incoming classes per stage, ELFS outperforms other sampling methods such as FreeSel (best label-free on CIFAR10) and random sampling under different storage budgets (10%, 20%, and 30%).

### C.6 DIFFERENCE BETWEEN LABEL-FREE CORESET SELECTION AND SEMI-SUPERVISED LEARNING

Semi-supervised learning (SSL) (Wang et al., 2022; Li et al., 2023) is another efficient way to address the challenge of high labeling costs in machine learning. SSL leverages a small, randomly sampled labeled dataset, often as little as 0.2% or even 0.02% of the full dataset (Wang et al., 2022; Li et al., 2023), along with a large pool of unlabeled data, to train models more effectively. However, the output of SSL and coreset selection methods are fundamentally different. While SSL focuses on training a model by utilizing both labeled and unlabeled data, coreset selection outputs not only a trained model but *also a compact, high-quality labeled subset (the coreset)*. This selected coreset can be reused for a variety of downstream tasks, such as continual learning (Yoon et al., 2021; Borsos et al., 2020) or neural architecture search (NAS) (Shim et al., 2021).

Similar to Yoon et al. (2021), we conduct an additional evaluation of coreset-based methods in a class-incremental continual learning setting to demonstrate the effectiveness of ELFS in downstream tasks, as shown in Figure 9. In this experiment, the training process of CIFAR10 is divided into five stages, with two new classes introduced incrementally at each stage. At every stage, a fixed coreset budget (10%, 20%, or 30%) is used to train a model. The results demonstrate that ELFS consistently outperforms other sampling methods, including FreeSel (the best-performing label-free method on CIFAR10) and random sampling, across all storage budgets. These findings highlight the robustness and efficiency of ELFS in preserving performance under constrained memory while adapting to new data incrementally. This emphasizes that ELFS is suitable for real-world continual learning scenarios with limited computational resources and memory.

## D ADDITIONAL DISCUSSION ON PRUNING HARD EXAMPLES AND GRID SEARCH ON HARD PRUNING RATE

As discussed in Section 4.3.2, pruning hard example helps improve the selected coreset quality. In this section, we will discuss how pruning hard examples help improve the quality of coreset and the overhead of identifying optimal $\beta$ to prune hard examples.

**The impact of hyperparameter $\beta$ in coreset selection.** Similar to previous work Zheng et al. (2023); Maharana et al. (2023) in this area, ELFS involves hard pruning rate $\beta$ as a hyperparameter, which aims to control the percentage of hard examples to be pruned. Higher $\beta$ means that more hard data is pruned. The reasoning that we prune hard examples is based on two insights: (1) Mislabeled examples often also have higher difficulty scores, which are harmful to training Pleiss et al. (2020), and (2) pruning hard data can allocate more budget to cover high-density areas Zheng et al. (2023). Grid searching $\beta$ is also a common practice in SOTA baselines Zheng et al. (2023); Maharana et al. (2023).

**Grid search overhead on $\beta$.** Grid searching $\beta$ is a common practice in SOTA coreset selection work Zheng et al. (2023); Maharana et al. (2023) and can introduce additional overhead for coreset selection. However, we argue that the overhead of the grid search is feasible, even for million-level

large-scale datasets. For instance, we evaluate ELFS on the ImageNet-1K dataset, which contains over one million images. The grid search time for a single pruning rate is approximately 17 hours using four A6000 GPUs. Given that our primary goal is to reduce human annotation costs, this computational overhead is relatively small compared to the significant savings in labeling efforts.

**More efficient $\beta$ selection.** One potential way to optimize the choice of $\beta$ is to leverage the convex relationship between beta and pseudo-label accuracy, as shown in Figure 5. This convexity allows us to stop searching once the accuracy starts to drop, which reduces the search space. However, we believe that there can be a more efficient and effective way to search the optimal $\beta$ and further enhance our method.

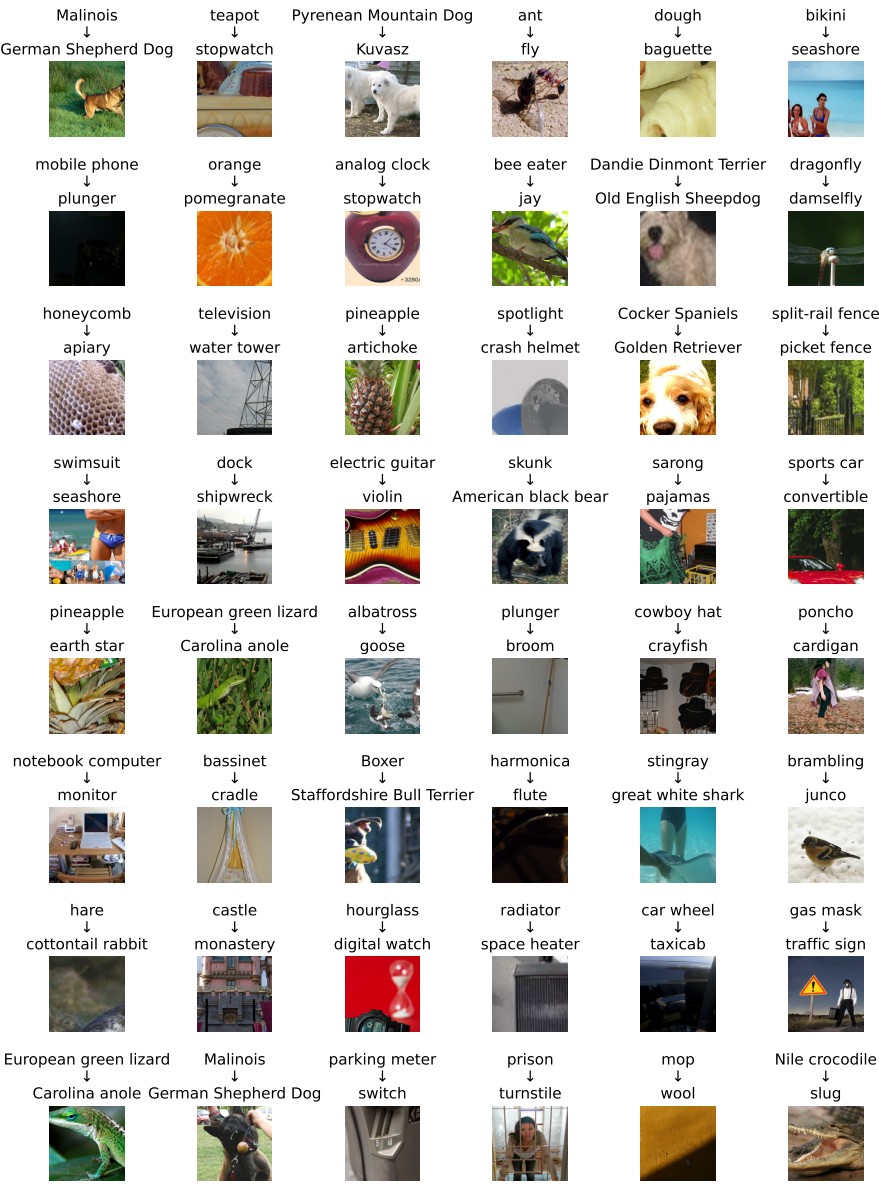

Figure 10: Randomly sampled ImageNet mislabeled examples. Each image features a caption that reads "Ground-Truth Label → DINO Pseudo-Label" at the top. The pseudo-labels are generated using the Hungarian match algorithm (Kuhn, 1955).

