# OpenReview forum: "ELFS: Label-Free Coreset Selection with Proxy Training Dynamics"
_ICLR.cc/2025/Conference — ICLR 2025 Poster_

### Official Review · Reviewer_u4aX · 2024-10-25

**Soundness:** 3
**Presentation:** 3
**Contribution:** 3
**Rating:** 8
**Confidence:** 3

**Summary:**

This paper proposes a new policy to sample a core subset for deep models. It introduces a deep clustering with the pseudo-labelling to estimate the score for each sample. Meanwhile, they try to fix the bias issue of pseudo-labelling. Experiments demonstrate the effectiveness of the proposed method.

**Strengths:**

1. The motivation of this paper is solid, and the topic of this paper exactly matches ICLR.
2. The writing of introduction clearly delivered the motivation and idea.
3. The experiment result looks good. It's interesting that many methods even cannot beat random sampling as suggested in Tab.1.
4. The ablation study is extensive.

**Weaknesses:**

1. Some sentences are redundant, such as these two questions proposed in the paper.
2. It would be better to move sec 4.1 to sec 3 to give readers an overview of the problem you are solving.
3. My **main concern** is that more benchmarks in different distributions should be evaluated. As described in the paper, this method relies on a pretrained vision encoder to get the visual features for each sample. Then, a deep clustering algorithm is introduced to get the pseudo labels and scores. However, the evaluated datasets in this paper are too easy for pretrained vision encoders. I believe that much of the data in the evaluation datasets is included during pretraining. If we use a dataset in a different distribution, such as a medical image dataset, without a good visual feature, will this method still work?

**Questions:**

1. Please explain why double-end pruning helps the performance.
2. Do you fine-tune the model with the coreset? Or do you train the model from scratch?
3. Do you use only the coreset to train the model? It would be better to show the result of using the coreset as the labelled set and the rest data as the unlabelled set to train a model with a semi-supervised learning algorithm such as SemiReward[1]. If, with the help of semi-supervised learning, a randomly sampled labelled set achieves good performance, and the labelled set selected by your model yields similar performance, then the benefits of using a coreset to train the model need to be clarified.
```
[1] SemiReward: A General Reward Model for Semi-supervised Learning, Siyuan Li and Weiyang Jin and Zedong Wang and Fang Wu and Zicheng Liu and Cheng Tan and Stan Z. Li, ICLR 2024
```

---

> ### Author Response · Authors · 2024-11-20
> **Response to Reviewer u4aX (1/2)**
>
> We are glad that Reviewer u4aX found our paper well-motivated, and our empirical results promising. We thank the reviewer for the constructive feedback. Below, we address the specific questions and concerns raised:
>
> > **Q1:** Some sentences are redundant, such as these two questions proposed in the paper.
>
> **A1:** We thank the reviewer for this suggestion, we have removed the redundant sentences in the revised version.
>
> > **Q2:** Move sec 4.1 to sec 3 to give readers an overview of the problem you are solving.
>
> **A2:** We thank the reviewer for this suggestion, we have moved Section 4.1 to Section 3 in our revised paper.
>
> > **Q3**:  My main concern is that more benchmarks in different distributions should be evaluated. As described in the paper, this method relies on a pretrained vision encoder to get the visual features for each sample. Then, a deep clustering algorithm is introduced to get the pseudo labels and scores. However, the evaluated datasets in this paper are too easy for pretrained vision encoders. I believe that much of the data in the evaluation datasets is included during pretraining. If we use a dataset in a different distribution, such as a medical image dataset, without a good visual feature, will this method still work?
>
> **A3**: We agree with the reviewer that the distribution mismatch between encoders and the targeted dataset can influence coreset selection performance. However, we would like to clarify that **ELFS does not always depend on a pretrained encoder.** As we discussed in Section 5.3.5, we study the case that a pretrained encoder is *unavailable*. We propose to use the targeted unlabeled dataset to train an encoder with unsupervised learning from scratch. The results in Table 6 indicate that even in this challenging setup, ELFS consistently outperforms other SOTA baselines that use pretrained encoders, further demonstrating the effectiveness of ELFS.
>
>
> > **Q4:** Please explain why double-end pruning helps the performance.
>
> **A4**:
> **Why does pruning hard examples improve the performance?**
>
> Similar to prior work [2,3], ELFS prunes hard examples to improve the quality of the selected coresets. As studied in [2], the reasoning behind pruning hard examples is based on two insights: (1) Mislabeled examples often also have higher difficulty scores, which are harmful to training [1], and (2) pruning hard examples allows for allocating more sampling budget to better cover high-density regions[2], which helps sample coreset providing better coverage on the underlying data distribution.
>
> **Why does double-end pruning improve the performance?**
>
> As discussed in Section 4.3.2 (revised version), we first directly apply a SOTA method CCS [2] to sample the coreset. However, pseudo-labels introduce inaccuracies, leading to a distribution shift in the calculated difficulty scores. Specifically, CCS, when using pseudo-label-based scores, tends to sample more easy examples than expected. To address this bias, instead of using stratified sampling after pruning hard examples as CCS does, we propose a double-end pruning approach to prune more easy examples than stratified sampling (as illustrated in Figure 4), which leads to better empirical performance.

---

> ### Author Response · Authors · 2024-11-20
> **Response to Reviewer u4aX (2/2)**
>
> > **Q5**: Do you fine-tune the model with the coreset? Or do you train the model from scratch?
>
> **A5**: We train the model from scratch. We have included this information in Section 5.1 in the revised paper.
>
> > **Q6**: Do you use only the coreset to train the model? It would be better to show the result of using the coreset as the labeled set and the rest data as the unlabelled set to train a model with a semi-supervised learning algorithm such as SemiReward. If, with the help of semi-supervised learning, a randomly sampled labeled set achieves good performance, and the labeled set selected by your model yields similar performance, then the benefits of using a coreset to train the model need to be clarified.
>
> **A6**:  Yes, we use only the coreset to train models.
>
> We thank the reviewer for bringing the connection between coreset selection and semi-supervised learning (SSL) for discussion. We agree with the reviewer that both SSL and label-free coreset selection can be used to reduce human-labeling efforts for training models. However, the goal and output of SSL and coreset selection methods are different. While SSL focuses on training a *model* by utilizing both labeled and unlabeled data, coreset selection aims to select *a compact, high-quality coreset* to represent the underlying distribution. Compared to SSL, label-free coreset selection also has many other downstream applications, like continual learning [5] or neural architecture search (NAS) [6], which provide unique values for label-free coreset selection.
>
> **Continual Learning Experiment**.
> To better support our argument, we conduct an evaluation of the coreset application on class incremental continual learning [4] on CIFAR10. Specifically, the training phase is split into 5 stages, with 2 new classes introduced incrementally at each stage.
> At every stage, a fixed budget (10%, 20%, or 30%) is labeled for model training.
> We report the results in $\textcolor{blue}{\text{Figure 9 in Appendix C.6}}$ of the revised paper. The evaluation results show that ELFS consistently outperforms baselines like random sampling and FreeSel (the best-performing label-free baseline on CIFAR10).
>
> We have included the discussion and evaluation in $\textcolor{blue}{\text{Appendix C.6}}$ of the revised paper.
>
> ---
>
>
> We sincerely thank Reviewer u4aX for the insightful feedback and comments, which inherently help improve the quality of our paper. We hope that our response and revisions have adequately addressed your concerns and questions. We are more than glad to answer any further questions.
>
> ---
>
>
> [1] Pleiss, Geoff, et al. "Identifying mislabeled data using the area under the margin ranking." NeurIPS 2020.
>
> [2] Zheng, Haizhong, et al. “Coverage-centric coreset selection for high pruning rates”. ICLR 2023.
>
> [3] Maharana, Adyasha, et al. “D2 pruning: Message passing for balancing diversity and difficulty in data pruning”. ICLR 2024.
>
> [4] Zhou, Da-Wei, et al. "Class-incremental learning: A survey." TPAML 2024
>
> [5] Yoon, Jaehong, et al. “Online Coreset Selection for Rehearsal-based Continual Learning”. ICLR 2022.
>
> [6] Shim, Jae-hun, et al. “Core-set Sampling for Efficient Neural Architecture Search“, ICML 2021 Workshop on Subset Selection in ML

---

> ### Comment · Reviewer_u4aX · 2024-11-22
>
> Thanks for your reply. I will increase my rating.

---

> > ### Author Response · Authors · 2024-11-22
> > **Thanks for the response**
> >
> > We thank the reviewer for the response and for raising the score. We are glad to see that our responses have addressed your concerns!

---

### Official Review · Reviewer_NwUk · 2024-10-30

**Soundness:** 3
**Presentation:** 2
**Contribution:** 3
**Rating:** 6
**Confidence:** 2

**Summary:**

The paper introduces a novel method called ELFS (Effective Label-Free Coreset Selection) for selecting coresets without relying on labeled data. This approach uses pseudo-labels derived from deep clustering to approximate training dynamics, enabling the estimation of data difficulty scores. These scores help identify coresets that can be labeled for training high-performance models while minimizing human annotation costs. ELFS addresses the significant performance gap typically found in label-free coreset selection by introducing a double-end pruning technique to manage the distribution shift caused by pseudo-label inaccuracies. This method shows notable improvements in various vision benchmarks over existing label-free methods, demonstrating its ability to approximate the effectiveness of supervised coreset selection.

**Strengths:**

ELFS presents a compelling label-free coreset selection method that reduces the need for extensive and costly labeled datasets while achieving accuracy close to supervised methods. By effectively utilizing pseudo-labels, ELFS not only significantly outperforms other label-free baselines but also exhibits strong performance despite the inherent inaccuracies and noise associated with pseudo-labels. Moreover, the method demonstrates robustness and versatility, showing good transferability across different datasets and model architectures, thereby enhancing its applicability in diverse machine learning tasks.

**Weaknesses:**

The ELFS method is quite effective, but it mainly builds on familiar techniques like pseudo-labeling and coreset selection. This might make it seem less novel or groundbreaking to those familiar with the field. Despite this, it does a great job using these methods to ensure high accuracy and reliability.

Moreover, to really show how well ELFS works and to expand its use, it would be beneficial to test it on a wider variety of datasets. This includes tackling larger and more complex datasets such as ImageNet, as well as datasets with uneven distributions or long tails. Testing ELFS in these contexts would help validate its effectiveness across different challenges and environments.

Potential Application Areas for ELFS: Beyond vision tasks, are there other types of data or tasks where ELFS could be effectively applied? Exploring its adaptability to different domains like text, audio, or even structured data could open up new applications.

Explanation of Hard and Easy Examples in Section 4.4.2: Could a visual representation or graph be used to clarify the difference between hard and easy examples as discussed in the section? Visual aids could help illustrate how ELFS handles these types of data, enhancing understanding of its approach.

Analysis of Data Distribution in Table 1: Is it possible to analyze further how the data distribution of the coreset selected by Random compares to that selected by ELFS? Understanding the differences in selection criteria and resulting coreset characteristics could provide deeper insights into the strengths and limitations of ELFS compared to simpler random sampling methods.

**Questions:**

See weaknesses.

---

> ### Author Response · Authors · 2024-11-20
> **Response to Reviewer NwUk (1/2)**
>
> We are glad that Reviewer NwUk found our method effective and compelling. We thank the reviewer for the constructive feedback. Below, we address the specific questions and concerns raised:
>
>
> > **Q1:** The ELFS method is quite effective, but it mainly builds on familiar techniques like pseudo-labeling and coreset selection. This might make it seem less novel or groundbreaking to those familiar with the field. Despite this, it does a great job using these methods to ensure high accuracy and reliability.
>
> **A1:** We thank the reviewer for acknowledging that our work does a great job of ensuring high accuracy and reliability. Regarding the concern about novelty, we would like to clarify that our proposed method, ELFS, does not merely apply existing techniques but has two unique contributions to achieve good performance in label-free coreset selection: 1) As far as we know, we are the first work to show that clustering-based pseudo-labels can serve as a good proxy for difficulty score calculation. 2) To address this distribution shift of scores introduced by pseudo-labels, we propose a double-end pruning sampling method, substantially improving the coreset performance (Section 3.4 in the paper).
>
> > **Q2:** Moreover, to really show how well ELFS works and to expand its use, it would be beneficial to test it on a wider variety of datasets. This includes tackling larger and more complex datasets such as ImageNet, as well as datasets with uneven distributions or long tails. Testing ELFS in these contexts would help validate its effectiveness across different challenges and environments.
>
> **A2**: We thank the reviewer for suggesting that an evaluation on a complex dataset like ImageNet-1K can better support the effectiveness of our method. We have already included this result in Table 1 in the paper, and the result shows that ELFS consistently outperforms other baselines on ImageNet-1k. Additionally, all benchmarks used in our paper are widely adopted in recent SOTA works [1, 2, 3, 4], ensuring a comprehensive and fair evaluation among different baselines.
>
>
> > **Q3:** Potential Application Areas for ELFS: Beyond vision tasks, are there other types of data or tasks where ELFS could be effectively applied? Exploring its adaptability to different domains like text, audio, or even structured data could open up new applications.
>
> **A3**: Similar to other baselines [1,2,3,4], our work primarily focuses on the vision modality. However, we are optimistic that ELFS can be extended to other modalities. If an encoder for the target modality can effectively extract features and generate high-quality pseudo-labels, the ELFS pipeline can be applied to select coresets in those modalities as well. Exploring potential applications in other modalities is an interesting direction, and we plan to address it in future work.
>
>
> > **Q4**: Explanation of Hard and Easy Examples in Section 4.4.2: Could a visual representation or graph be used to clarify the difference between hard and easy examples as discussed in the section? Visual aids could help illustrate how ELFS handles these types of data, enhancing understanding of its approach.
>
> **A4**:  Thank you for your suggestion. We have added $\textcolor{blue}{\text{Figure 7}}$ and the corresponding analysis in $\textcolor{blue}{\text{Appendix C.4}}$ to provide a better illustration of hard-to-learn and easy-to-learn examples in the revised paper.
> In $\textcolor{blue}{\text{Figure 7}}$, we illustrate how AUM metric differentiates between **hard-to-learn examples** and **easy-to-learn examples**. **Hard-to-learn examples** ($\textcolor{blue}{\text{Figure 7a}}$) are characterized by fluctuating logits and a small margin between the green line (correct label) and the red line (largest incorrect label), reflecting the model's struggle to confidently predict the correct label. These examples often have ambiguous or noisy features. In contrast, **easy-to-learn examples** ($\textcolor{blue}{\text{Figure 7b}}$) exhibit a clear and consistent margin, with the green line dominating, indicating the model quickly learns these examples with confidence.

---

> ### Author Response · Authors · 2024-11-20
> **Response to Reviewer NwUk (2/2)**
>
> > **Q5**: Analysis of Data Distribution in Table 1: Is it possible to analyze further how the data distribution of the coreset selected by Random compares to that selected by ELFS? Understanding the differences in selection criteria and resulting coreset characteristics could provide deeper insights into the strengths and limitations of ELFS compared to simpler random sampling methods.
>
> **A5**:
> Thank you for your suggestion. We have added $\textcolor{blue}{\text{Figure 8}}$ and the corresponding analysis in $\textcolor{blue}{\text{Appendix C.5}}$ to provide a deeper analysis of how ELFS compares to random sampling in selecting coresets in the revised paper.
> $\textcolor{blue}{\text{Figure 8}}$ analyzes the data distribution of the coresets selected by ELFS and random sampling on CIFAR100 with a 50% pruning rate. The graph shows that random sampling selects more easy-to-learn examples due to their higher density in the dataset, as shown by the concentration of green bars at higher AUM values. However, this results in fewer hard-to-learn examples—data points with lower AUM values—that are critical for improving the model's robustness. In contrast, ELFS selects a more balanced coreset with better data difficulty diversity. This difference explains the performance gap between the two methods, with ELFS achieving 73.6% compared to random sampling’s 71.1%.
>
> ---
>
> We sincerely thank Reviewer NwUk for the insightful feedback and comments, which inherently help improve the quality of our paper. We hope that our response and revisions have adequately addressed your concerns and questions. We are more than glad to answer any further questions.
>
> ---
>
>
>
> [1] Zheng, Haizhong, et al. “Coverage-centric coreset selection for high pruning rates”. ICLR 2023.
>
> [2] Maharana, Adyasha, et al. “D2 pruning: Message passing for balancing diversity and difficulty in data pruning”. ICLR 2024
>
> [3] Xia, Xiaobo, et al. “Moderate Coreset: A Universal Method of Data Selection for Real-world Data-efficient Deep Learning”, ICLR 2023.
>
> [4] Sorscher, Ben, et al. “Beyond neural scaling laws: beating power law scaling via data pruning”. NeurIPS 2022

---

### Official Review · Reviewer_ChTP · 2024-10-31

**Soundness:** 3
**Presentation:** 3
**Contribution:** 2
**Rating:** 6
**Confidence:** 4

**Summary:**

This paper proposes a new label-free coreset selection algorithm called ELFS to relieve the costly human annotation efforts. ELFS utilizes the deep clustering to generate pseudo-labels and estimate data difficulty scores. Afterwards, a double-end pruning method is introduced to mitigate the bias of data difficulty scores. Experiments show that ELFS can surpass previous label-free coreset selection baselines on several benchmarks.

**Strengths:**

1. It is an elegant and effective idea to estimate the data difficulty score through deep clustering. This handles the challenge to measure the prediction uncertainty and sample difficulty without any human labels.

2. The proposed method is evaluated on multiple classification benchmark, showing notable performance gain compared with state-of-the-arts.The design of each module is well justified through ablation studies.

**Weaknesses:**

1. My major concern lies in the selection of hyper-parameter $\beta$. I can understand they require some grid search for hyper-parameters. However, according to Fig. 5, the optimal value is different for multiple datasets or sampling ratios, which is quite inefficient. For example, if there is a large dataset with millions of images, it is infeasible to do grid search on it.

2. Based on Tab. 7, it is quite strange that ResNet50 cannot outperform ResNet18 on the selected subset. I assume it reasons from the simplicity of CIFAR10. Maybe the authors can do the transferability experiments on complex datasets like ImageNet since it is a main difference between corset selection and active learning.

3. For Sec. 4.1, I assume the formulation of label-free coreset selection is already covered in previous work. It may be moved to Sec. 3 for clarity.

**Questions:**

Please consider responding to the weaknesses.

---

> ### Author Response · Authors · 2024-11-20
> **Response to Reviewer ChTP**
>
> We are glad that Reviewer ChTP found our method elegant and effective. We thank the reviewer for the constructive feedback. Below, we address the specific questions and concerns raised:
>
>
> >**Q1:** My major concern lies in the selection of hyper-parameter $\beta$. I can understand they require some grid search for hyper-parameters. However, according to Fig. 5, the optimal value is different for multiple datasets or sampling ratios, which is quite inefficient. For example, if there is a large dataset with millions of images, it is infeasible to do grid search on it.
>
>
> **A1:** We thank the reviewer for bringing the computational overhead of hyperparameter $\beta$ search for discussion. We acknowledge that grid search $\beta$ can introduce additional overhead for coreset selection. However, we argue that this $\beta$ grid search strategy is a common practice in SOTA coreset selection work [1,2], as pruning hard examples increases the coreset quality. Moreover, this grid search is feasible even for million-level large-scale datasets. For instance, in Section 5.2, we evaluate ELFS on the ImageNet-1K dataset, which contains over one million images.  The grid search time for a single pruning rate is approximately 17 hours using four A6000 GPUs. Given that our primary goal is to reduce human annotation costs, this computational overhead is relatively small compared to the significant savings in labeling efforts.
>
>
> However, we agree with the reviewer that improving the efficiency of the hard-pruning rate $\beta$ search is an important direction, and we plan to explore a more efficient hyperparameter search in our future work.
>
>
> We have also included this additional discussion in Appendix.D in the revised paper for better clarification.
>
>
>
>
> >**Q2:** Based on Tab. 7, it is quite strange that ResNet50 cannot outperform ResNet18 on the selected subset. I assume it reasons from the simplicity of CIFAR10. Maybe the authors can do the transferability experiments on complex datasets like ImageNet since it is a main difference between corset selection and active learning.
>
>
> **A2**: We thank the reviewer for suggesting this experiment. We agree that the slight drop in ResNet-50 can be due to the simplicity of CIFAR10. As suggested by the reviewer, we conducted the transferability experiment on ImageNet-1K (shown in the below table). The coreset is selected on ResNet-34 architecture and then transferred to train ResNet-50. The results show that the selected coresets exhibit good transferability on ResNet-50, and ResNet-50 achieves better accuracy than ResNet-34. We have added the results to $\textcolor{blue}{\text{Table 12 in Appendix C.3}}$ in the revised version for better comparison.
>
>
> **Table: ImageNet-1K ResNet-50 Transferability Ablation Study.**
> | Pruning Rate | 30% | 50% | 70% | 80% | 90% |
> |--------------|------|------|------|------|------|
> | ResNet 34 | 73.5 | 71.8 | 67.2 | 63.4 | 54.9 |
> | ResNet 50 | 76.2 | 74.0 | 69.5 | 64.6 | 56.4 |
>
>
>
>
>
>
> >**Q3**: For Sec. 4.1, I assume the formulation of label-free coreset selection is already covered in previous work. It may be moved to Sec. 3 for clarity.
>
> **A3**: We thank the reviewer for this suggestion, we have moved Section 4.1 to Section 3 in our revised paper.
>
> ---
>
> We sincerely thank Reviewer ChTP for the insightful feedback and comments, which inherently help improve the quality of our paper. We hope that our response and revisions have adequately addressed your concerns and questions. We are more than glad to answer any further questions.
>
> ---
>
> [1] Zheng, Haizhong, et al. “Coverage-centric coreset selection for high pruning rates”. ICLR 2023.
>
> [2] Maharana, Adyasha, et al. “D2 pruning: Message passing for balancing diversity and difficulty in data pruning”. ICLR 2024

---

> ### Comment · Reviewer_ChTP · 2024-11-22
>
> Thanks for the detailed response from the authors. This helps to solve my concerns, so I increase my rating correspondingly.

---

> > ### Author Response · Authors · 2024-11-22
> > **Thanks for the response**
> >
> > We thank the reviewer for the response and for raising the score. We are glad to see that our responses have addressed your concerns!

---

### Official Review · Reviewer_sVc5 · 2024-11-02

**Soundness:** 4
**Presentation:** 4
**Contribution:** 3
**Rating:** 8
**Confidence:** 3

**Summary:**

The paper presents ELFS (Effective Label-Free Coreset Selection), a method designed to improve label-free coreset selection by estimating data difficulty scores without requiring ground truth labels. The authors tackle challenges in label-free selection by employing pseudo-labels from deep clustering to approximate training dynamics and mitigate distribution shifts with a double-end pruning technique. ELFS shows superior performance over existing label-free methods across various vision benchmarks (e.g., CIFAR10, CIFAR100, and ImageNet-1K) and achieves results close to those of supervised selection methods.

**Strengths:**

1. ELFS effectively addresses the limitations of previous label-free coreset selection approaches, providing a feasible solution that leverages deep clustering for pseudo-labeling.

2. By employing double-end pruning, ELFS improves the selection of informative samples, achieving consistent performance improvements over baselines, even in challenging scenarios.

3. The evaluation across multiple datasets and pruning rates, along with an ablation study, showcases ELFS's flexibility and robustness, which may benefit a range of vision tasks.

4. The authors show that including more challenging samples enhances model performance, with ELFS effectively prioritizing hard examples through double-end pruning.

**Weaknesses:**

1. The experiments involve numerous hyperparameters, optimized through grid search. A more in-depth analysis of the underlying reasons behind these optimal values would strengthen the understanding of how different parameters affect the measurement of sample difficulty, offering clearer insights into the importance of hard examples.

2. The approach heavily relies on feature extractors like SwAV and DINO for clustering. It remains unclear if using more advanced encoders, such as CLIP, could further improve performance or stability, suggesting potential limits in ELFS's generalizability with different encoders.

**Questions:**

1. Given the grid search used to determine optimal hyperparameters, could a deeper analysis reveal why certain values work best for measuring sample difficulty? Specifically, how do these parameters influence the balance between easy and hard examples selected for the coreset, and could this inform a more consistent method for tuning them?

2. ELFS currently uses SwAV and DINO as feature extractors for clustering. Would more powerful encoders, such as CLIP, improve the quality of pseudo-labels or provide more stable performance across datasets? Additionally, what effect might these alternative encoders have on the distribution of selected hard and easy examples?

---

> ### Author Response · Authors · 2024-11-20
> **Response to Reviewer sVc5**
>
> We are glad that Reviewer sVc5 found our approach effective, and our evaluation and ablation study extensive. We thank the reviewer for the constructive feedback. Below, we address the specific questions and concerns raised:
>
> > **Q1:** The experiments involve numerous hyperparameters, optimized through grid search. A more in-depth analysis of the underlying reasons behind these optimal values would strengthen the understanding of how different parameters affect the measurement of sample difficulty, offering clearer insights into the importance of hard examples.
> >
> > Given the grid search used to determine optimal hyperparameters, could a deeper analysis reveal why certain values work best for measuring sample difficulty? Specifically, how do these parameters influence the balance between easy and hard examples selected for the coreset, and could this inform a more consistent method for tuning them?
>
> **A1**: We thank the reviewer for bringing the hyperparameter hard pruning rate for discussion.
>
> **How does the hyperparameter impact the training?**
> Similar to previous work [2,3] in this area, ELFS involves hard pruning rate $\beta$ as a hyperparameter, which aims to control the percentage of hard examples to be pruned. Higher $\beta$ means that more hard data is pruned. The reasoning that we prune hard examples is based on two insights: 1) Mislabeled examples often also have higher difficulty scores, which are harmful to training [1], and 2) pruning hard data can allocate more budget to cover high-density areas [2]. Grid searching $\beta$ is also a common practice in SOTA baselines [2,3].
>
> **Can we have a more efficient and consistent method for tuning $\beta$?**
> We agree with the reviewer that a more efficient search on $\beta$ can improve the efficiency of our method. One potential way to optimize the choice of beta is to leverage the convex relationship between beta and pseudo-label accuracy, as shown in Figure 5. This convexity allows us to stop searching once the accuracy starts to drop, which reduces the search space. However, we believe that there can be a more efficient and effective way to search the optimal $\beta$ and further enhance our method. We plan to leave this exploration for future study.
>
> We have included those discussions in $\textcolor{blue}{\text{Appendix D}}$ in the revised paper.
>
> > **Q2**: The approach heavily relies on feature extractors like SwAV and DINO for clustering. It remains unclear if using more advanced encoders, such as CLIP, could further improve performance or stability, suggesting potential limits in ELFS's generalizability with different encoders.
> >
> >ELFS currently uses SwAV and DINO as feature extractors for clustering. Would more powerful encoders, such as CLIP, improve the quality of pseudo-labels or provide more stable performance across datasets? Additionally, what effect might these alternative encoders have on the distribution of selected hard and easy examples?
>
> **A2**:  **Evaluation results on CLIP**. As suggested by the reviewer, we conducted additional experiments using CLIP as the encoder on CIFAR10/100 (shown in the tables below). Our results show that the quality pseudo-labels generated by CLIP are better than SwAv and worse than DINO on the CIFAR dataset (Note that DINO is released after CLIP).
> We have a similar ablation study in Section 5.3.1 and Section 5.3.4: a better encoder can provide higher-quality pseudo-labels, and therefore improve the quality of the coreset selected.
>
> ---
> **Table: Additional evaluation using CLIP on CIFAR10**
> | Encoder | Pseudo-Label Acc | 30%  | 50%  | 70%  | 80%  | 90%  |
> |-|-|-|-|-|-|-|
>  | DINO | 92.5%  | 95.5 | 95.2 | 93.2 | 90.7 | 87.3 |
>  | CLIP| 85.2%  | 95.0 | 94.5 | 92.7 | 90.1 | 84.5 |
>  | SwAv | 60.7% | 95.0 | 94.3 | 91.8 | 89.8 | 82.5 |
> | Best Label-Free|  | 94.7 | 93.8 | 91.7 | 88.9 | 82.4 |
>
> ---
>
> **Table: Additional evaluation using CLIP on CIFAR100**
> | Encoder | Pseudo-Label Acc  | 30%  | 50%  | 70%  | 80%  | 90%  |
> |-|-|-|-|-|-|-|
> | DINO  | 66.3%     | 76.8 | 73.6 | 68.4 | 62.3 | 54.9 |
> | CLIP   | 54.0%         | 75.5 | 73.1 | 66.5 | 58.4 | 50.6 |
> | SwAv  | 39.8%             | 76.1 | 72.1 | 65.5 | 58.2 | 49.8 |
> |Best Label-Free  |    |  75.3 | 71.8 | 66.0  | 58.9  | 44.8 |
>
>
> We also included the CLIP results in $\textcolor{blue}{\text{Table 11 in Appendix C.2}}$ in the revised paper.
>
> ---
>
> We sincerely thank Reviewer sVc5 for the insightful feedback and comments, which inherently help improve the quality of our paper. We hope that our response and revisions have adequately addressed your concerns and questions. We are more than glad to answer any further questions.
>
> ---
>
>
> [1] Pleiss, Geoff, et al. "Identifying mislabeled data using the area under the margin ranking." NeurIPS 2020.
>
> [2] Zheng, Haizhong, et al. “Coverage-centric coreset selection for high pruning rates”. ICLR 2023.
>
> [3] Maharana, Adyasha, et al. “D2 pruning: Message passing for balancing diversity and difficulty in data pruning”. ICLR 2024

---

### Author Response · Authors · 2024-11-20
**General Response**

Dear all reviewers,

We sincerely thank all the reviewers for their constructive reviews and insightful feedback. These comments and suggestions have been very helpful in improving the quality of our paper.

We are pleased that reviewers found our method ELFS showing “notable performance gain” (All reviewers); “extensive evaluation and ablation study” (sVc5, ChTP, u4aX); “elegant and effective” (ChTP); and “robust and versatile” (NwUk). We appreciate the recognition by all reviewers of our method's effectiveness and generalizability.

For the comments and concerns discussed in the reviews, we write separate responses to each individual review.  Below we summarize the revisions we made to the paper. We also highlight all modifications in blue in the revised paper to make the comparison more clear.


## Paper Revision Summary:

**Additional evaluation and analysis:**

1. $\textcolor{blue}{\text{Table 11 in Appendix C.2}}$: Evaluation using CLIP for better ablation study on encoders; (suggested by Reviewer sVc5)

2. $\textcolor{blue}{\text{Table 12 in Appendix C.3}}$: Transferability evaluation on ImageNet-1K; (suggested by Reviewer ChTP)

3. $\textcolor{blue}{\text{Figure 7 in Appendix C.4}}$: Case study visualization of easy-to-learn and hard-to-learn data selected by AUM; (suggested by Reviewer NwUk)

4. $\textcolor{blue}{\text{Figure 8 in Appendix C.5}}$: Data distribution comparison between ELFS and random sampling; (suggested by Reviewer NwUk)

5. $\textcolor{blue}{\text{Figure 9 in Appendix C.6}}$: Coreset-based continual learning evaluation. (suggested by Reviewer u4aX)

**Writing:**

1. We moved $\textcolor{blue}{\text{Section 4.1}}$ to $\textcolor{blue}{\text{Section 3.1}}$ for a better problem statement; (suggested by Reviewer ChTP and u4aX)

2. Removed the redundant sentences to improve the readability; (suggested by Reviewer u4aX)

3. Added training setting details in $\textcolor{blue}{\text{Section 5}}$; (suggested by Reviewer u4aX)

4. Added discussion on the connection and difference between semi-supervised learning and label-free coreset selection in $\textcolor{blue}{\text{Appendix C.6}}$; (suggested by Reviewer u4aX)

5. Added discussion on hard data pruning and grid search overhead in $\textcolor{blue}{\text{Appendix D}}$. (suggested by Reviewer sVc5 and ChTP)

---

We thank all the reviewers for their constructive feedback that has helped us improve the paper. We hope that our responses and the revised paper addressed the reviewers’ concerns and are happy to answer any further questions.

---

### Meta-Review · Area_Chair_9Pts · 2024-12-21

**Metareview:**

This paper proposes label-free coreset selection by using deep clustering to estimate data difficulty scores w/o gt labels and this paper proposes a double-end pruning method to mitigate bias on these calculated scores. After carefully considering the reviewers' feedback and the author's rebuttal, the Area Chair (AC) agrees with the positive assessment provided by the four reviewers. All reviewers have reached a consensus to accept the paper, recognizing its merits. The AC has thoroughly reviewed the comments and rebuttal and concurs with the overall evaluation. Consequently, the AC recommends accepting this paper for publication.

**Additional Comments On Reviewer Discussion:**

The main concerns from reviews are summarized as follows:
- Hyperparameter tuning for training.
- Evaluations using different encoders.
- Experiments on ImageNet-1k.
- Application areas of the proposed method.
- Explanations of hard and easy examples.
- Analysis of data distribution in Table 1.
- Why double-end pruning helps.
-  Train the model from scratch or not.
- Only the coreset is used to train the model.

The rebuttal has well addressed these concerns point-by-point with updated experiments and careful explanations, and AC agrees with these discussions and changes.

---

### Decision · Program_Chairs · 2025-01-22

Accept (Poster)